# Insights on the Regeneration Potential of Müller Glia in the Mammalian Retina

**DOI:** 10.3390/cells10081957

**Published:** 2021-07-31

**Authors:** Ahmed Salman, Michelle E. McClements, Robert E. MacLaren

**Affiliations:** 1Nuffield Department of Clinical Neuroscience, University of Oxford, Oxford OX3 9DU, UK; michelle.mcclements@eye.ox.ac.uk (M.E.M.); robert.maclaren@eye.ox.ac.uk (R.E.M.); 2Oxford Eye Hospital, Oxford OX3 9DU, UK

**Keywords:** Müller glia, reprogramming, retinal regeneration, regeneration potential, stem cells, proliferation, differentiation

## Abstract

Müller glia, the major glial cell types in the retina, maintain retinal homeostasis and provide structural support to retinal photoreceptors. They also possess regenerative potential that might be used for retinal repair in response to injury or disease. In teleost fish (such as zebrafish), the Müller glia response to injury involves reprogramming events that result in a population of proliferative neural progenitors that can regenerate the injured retina. Recent studies have revealed several important mechanisms for the regenerative capacity of Müller glia in fish, which may shed more light on the mechanisms of Müller glia reprogramming and regeneration in mammals. Mammalian Müller glia can adopt stem cell characteristics, and in response to special conditions, be persuaded to proliferate and regenerate, although their native regeneration potential is limited. In this review, we consider the work to date revealing the regenerative potential of the mammalian Müller glia and discuss whether they are a potential source for cell regeneration therapy in humans.

## 1. Introduction

Vision is often reported as the sense most adults (88%) would least like to lose [1]. Blindness can result from injury or disease, with retinal disease caused by loss or damage to the highly organised laminar structure of specialised neural cells at the back of the eye. Diseases such as retinitis pigmentosa, age-related macular degeneration, diabetic retinopathy and glaucoma can lead to blindness due to the loss of function of one or more of these specialised cell types. There are numerous strategies for attempting to prevent or delay sight loss and even to restore vision in those with no light perception. The treatment approach taken will depend on the stage and origin of sight loss, but current strategies include cell transplantation [2], gene supplementation [3], gene correction [4], optogenetics [5] and prosthetic devices [6]. Whilst these strategies are showing promising signs of safety and efficacy and are rapidly evolving and improving, they require surgical intervention for the delivery of viral vectors and/or reprogrammed cells and implants. A less invasive strategy for retinal regeneration could be for remaining cell types of the retina to initiate repair.

The idea of a self-renewing retina might seem unfeasible, but it is not unprecedented; zebrafish show a remarkable ability to regenerate the retina after injury [7,8,9]. This regeneration potential stems from one cell type in the retina, the Müller glia, which are astrocyte-like radial cells that play a pivotal role in maintaining retinal structure and homeostasis and are considered to be the main glial cells in the retina. In zebrafish, Müller glia undergo reprogramming events in response to injury and acquire stem cell characteristics and proliferate to progenitor cells, which leads to retinal repair [10,11,12,13,14,15]. Although this regeneration potential is common for all vertebrates, it is, however, absent in mammalian systems for unknown reasons. Gaining more understanding of the mechanisms of the regeneration potential in fish may help unlock the reasons why mammalian cells lack this phenomenon and how we might encourage it.

This review will consider the responses of Müller glia to retinal injury in mammals and the signalling mechanisms underlying their reprogramming and regeneration potential.

## 2. Müller Glia Anatomy and Function

The function of the retina is to convey light signals to the brain. The photoreceptor cells respond to light, triggering a phototransduction cascade that passes through the bipolar cells, then on to the retinal ganglion cells that converge into the optic nerve, which carries the signal to the brain. Horizontal cells enable interconnections of multiple photoreceptors and bipolar cells, whilst amacrine cells are the interneurons between bipolar cells and ganglion cells (Figure 1). These cell types are critical for the conductance of light signals to the brain but the neural retina also contains three types of glial cells: microglia and two types of macroglia, astrocytes and Müller cells, with the latter considered the main glial cell type in the retina [16]. These glial cells provide structure and functional support to the retina.

Müller glia are radial shaped cells that span the entire thickness of the neural retina from the outer to the inner limiting membranes (OLM and ILM, respectively) (Figure 1). They share a multipotent origin with neurones in the retinal neuroepithelium [17]. The process of originating Müller glia is not well understood but involves several signalling pathways such as Notch, retinal homeobox protein Rx (RAX) and Janus kinase (JAK) signalling [18,19,20,21,22]. Müller glia are developed at later stages of retinal development as revealed by ^3^H-Thymidine labelling studies [23]. The retinal ganglion cells are the first to be born, followed shortly by the horizontal, cone photoreceptors and displaced amacrine cells. At later stages, the amacrine, bipolar and Müller cells appear.

Müller cell processes make contact with all retinal neurones and blood vessels supporting the overall retinal structure and monitor homeostasis [24]. Their somata lie in the inner nuclear layer (INL) of the retina with processes that radiate from the soma in opposite directions, spanning the thickness of the entire retina and making contact with both the inner limiting membrane (ILM) and outer limiting membrane (OLM), where they form tight junctions with the photoreceptor layer [16]. Their lateral processes expand into the inner and outer plexiform layers (IPL and OPL, respectively), where they make sheaths around the synapses in these layers.

The unique anatomy of Müller glia means they are well-positioned to monitor retinal homeostasis and contribute to the overall structure of the retina. They provide homeostatic, metabolic and structural support to retinal neurones, regulate ion and water homeostasis and maintain the blood–retinal barrier [24]. Müller cells also impact the synaptic activity in the retina by recycling neurotransmitters including glutamate and gamma-Aminobutyric acid (GABA) [25,26,27]. The uptake of the neurotransmitter glutamate by the glutamate–aspartate transporter (GLAST) is known to be important in regulating levels in the inner and outer plexiform layers. This prevents the spread of glutamate beyond the synapses, which ensures a fine resolution and prevents glutamate toxicity [25,28]. Furthermore, GABA is converted to glutamate after uptake by Müller cells and is then converted to glutamine by a glutamine synthase, which is localised specifically in Müller cells [25]. The neurotransmitter recycling process continues with glutamine released by Müller glia to be taken up by neurones as a precursor to producing GABA and glutamate. Müller glia’s involvement in regulating the synaptic connectivity in the inner retina has been confirmed by both pharmacological blockade and downregulation of glutamine synthase, resulting in neuronal dysfunction in the retina [25,29].

Functions of Müller glia therefore impact, directly or indirectly, the neuronal activity of the retina. These cells also control the release of potassium ions across the intra- and extracellular spaces in the retina, preventing neuronal hyperexcitation that can be caused by an excess release of potassium ions [16].

The clearance of water from retinal tissue also appears to be mediated by Müller cells. The uptake of glucose, by the retinal epithelial cells, causes water influx from the blood that accumulates in the retina, coupled with water production from the oxidative degradation of glucose. There is also pressure-induced water influx by the vitreous, all of which causes water accumulation in the retina. Müller glia mediate an osmotically driven water clearance through ionic shift across the transmembrane facilitated by aquaporin-4 channels [24,30], in particular potassium [31], which redistributes excess water out of the retina.

These functions are a snapshot of the roles undertaken by the Müller glia to provide a general understanding of the importance of these cells in maintaining the health and function of the retina. Other functions include the anti-oxidative support of photoreceptors and retinal neurones [32], removal of CO_2_ and regulation of the extracellular pH [33], maintenance of the blood–retinal barrier [34,35] and regulation of the retinal blood flow [36]. Further details of such functions can be found in comprehensive reviews of these cell types.

## 3. Müller Glia Response to Injury in Mammals

Unlike in zebrafish, birds and amphibians, where the retina has spontaneous regeneration potential driven by Müller glia to repair the retina upon injury or disease [7,8,9,37], in the mammalian retina such regeneration potential is limited. Thus, human retinal disease often leads to a permanent loss of vision and blindness. Although Müller glia in the mammalian retina harbour stem cell characteristics [38], they are unable to regenerate the retina despite showing the ability to acquire characteristics of retinal neurons in vitro when cultured with selective growth factors that induce differentiation [39,40,41]. This limited capacity of retinal repair in the mammalian retina is also evident in vivo, and it is currently unknown whether factors or molecules induced during development or disease are halting this regeneration capacity in Müller glia.

Several types of cells in the mammalian retina, one of which is Müller cells, have been shown to exhibit neurogenic potential, the others being cells of the ciliary margin and the retinal pigment epithelium (RPE) [42,43,44,45]. Müller cells in mammals are normally quiescent; however, they are activated upon stress or injury in a process known as “reactive gliosis”, where they change morphology and undergo a shift in expression profile, dedifferentiation, migration and proliferation [46,47]. Reactive gliosis is beneficial to neurones as it prevents glutamate neurotoxicity, as noted earlier, as well as releasing a range of factors that prevent cell death [47]. These events somehow resemble processes that occur during retinal regeneration and repair in fish and birds, but with an absence of neuronal regeneration and retinal repair [46,47]. Mammalian Müller cells are therefore able to undergo gliosis, but seem unable to differentiate into retinal neurons. There are number of key events during Müller glia reprogramming in zebrafish that represent a potential intervention window in the mammalian systems to stimulate neuronal regeneration (Figure 2).

The limited, or lack of, ability of Müller cells to differentiate into retinal neurones has been challenged several times. Evidence from in vitro and in vivo studies has shown that Müller cells can undergo reprogramming and produce new bipolar and photoreceptor cells in response to N-Methyl-D-aspartic acid (NMDA) [48] or amacrine cells after NDMA-induced damage and growth factor treatment [49]. Differentiation into retinal neurones and photoreceptors has also been shown after growth factor treatment [38,50,51,52,53]. However, damage by intense light exposure did not seem to promote Müller cell proliferation [54].

Whilst there are indications of the regeneration potential of Müller glia, the current data are inconclusive in the extent to which this occurs or indeed how to exploit it with the mechanisms still largely unknown. Neuronal explants present an environment where neuronal loss occurs spontaneously, which triggers reactive gliosis [55]. Müller cells in retinal explants are not quiescent anymore and dedifferentiate/reprogram to acquire stem cell characteristics and start proliferation and neural regeneration. When growth factors that trigger Müller cells’ activation are introduced, more than half of the cell population starts proliferating, breaking the quiescent state [55]. NMDA-induced retinal damage of the mouse retina in combination with Epidermal Growth Factor (EGF) treatment is one of the most potent methods to stimulate Müller glia proliferation [49]. The Müller glia proliferation stimulated by NMDA and EGF treatment involves mitogen-activated protein kinase (MAPK), bone morphogenesis protein (BMP), and phosphoinositide 3-kinase (PI3K) signalling pathways [56]. During the early stages of retinal development in the mammalian retina when Müller glia are proliferating, the EGF receptor (EGFR) is upregulated. However, when Müller glia stop proliferating and become quiescent, EGFR expression is downregulated and this decrease in expression is accompanied by the overexpression of transforming growth factor-beta (TGF-B) signalling, which appears to have a suppressive effect on the proliferation of Müller cells postnatally [57,58].

Interestingly, the subretinal delivery of non-toxic doses of glutamate has been shown to stimulate Müller glia proliferation and neural regeneration in vivo [59]. This suggests that, under the right conditions, Müller glia do have the potential to reprogram and regenerate for retinal repair.

During retinal reprogramming in fish, the upregulation of the Achaete-scute homolog 1 (*Ascl1*) gene has particular importance in the reprogramming of Müller cells into neural progenitors [7,8,9,60]. Interestingly, the mouse *Ascl1* was not upregulated in NMDA-induced damage of the retina [49], which might go some way towards explaining the poor regenerative capacity of Müller cells in mammals. However, the overexpression of *Ascl1* is correlated with retinal damage in young mice [55], suggesting there is a higher potential of Müller cell reprogramming in young mice, where Müller glia are still proliferating [55,56]. In addition, the overexpression of *Ascl1* in vitro was sufficient to activate neurogenic reprogramming in Müller glia. This was also observed ex vivo in retinal explants, in which the overexpression of *Ascl1* caused the de-differentiation of Müller glia and expression of the bipolar marker protein kinase alpha (PKCa) [61].

## 4. Current Understanding of the Regeneration Potential of Müller Glia in Mammals

Certain signalling pathways such as Wnt/B-catenin [51,52], sonic hedgehog [62], epidermal growth factor (EGF)-EGF receptor (EGFR), glutamate and *Ascl1*-dependent signalling pathways [61] (Table 1), are known to trigger proliferative events and neural regeneration by Müller glia in the injured mammalian retina.

Historically, the regenerative potential of retinal glial cells was investigated by MacLaren et al. [63], where it was shown that the glial scars caused by surgical lesions in humans are different than other scars in the nervous system, in which a regrowth and reconnection of regenerating axons was observed. Eventually, it has been revealed that the regrowth was in fact of retinal ganglion cells.

Regarding the role of *Ascl1* in mammalian Müller glia, evidence of chromatin modification has emerged, showing an increased capacity of neuronal regeneration of Müller cells after retinal injury when both *Ascl1* and histone deacetylase inhibitor (HDAC) were overexpressed [64]. This suggests epigenetic modifications might hold a key role in modulating Müller glia regeneration capacity. Just like in fish, Müller glia acquire stem cell characteristics and start expressing pluripotent genes upon activation [10].

Interestingly, the promoter of *Ascl1* and other pluripotent genes such as *LIN28, HB-EGF*, *SOX2* and *OCT4* maintain a hypomethylated state that makes them more accessible, indicating that Müller glia might be acting as endogenous stem cells. To further reiterate the role of epigenetic modification, it was recently shown that targeted destruction of polypyrimidine tract-binding proteins (PTBs), which are micro-RNA coding proteins, results in the dedifferentiation of Müller glia into retinal ganglion cells in vivo [64]. In this study, the *Ptbp1* mRNA was targeted by CRISPR technology using Cas-Rx recombinase and delivered to the retina via a viral vector. The differentiation of Müller glia to ganglion cells was then assessed and the visual function of mice was also examined to evaluate the replenishment of the ganglion cell population in the retina. Research on the inhibition of micro-RNAs such as miR-9 and miR-124, as well as transcription factors involved in epigenetic modifications such as HDAC and the repressor element1-silencing transcription factor (REST), is becoming increasingly interested in understanding Müller glia regeneration potential. Recently, it has been suggested that micro-RNAs, such as miR-25 and miR-124, play an important role in Müller glia neurogenic potential in mice by inducing the expression of *Asl1*, leading to the conversion of mature Müller glia into neural progenitor cells [73]. Another recent study has shown that the inhibition of miR-28 upregulates CRX, a photoreceptor-specific marker, which leads to the transdifferentiation of Müller glia progenitor cells into photoreceptors [74]. Understanding the nature of epigenetic modifications might reveal key mechanisms of the Müller glia regeneration potential, or the lack of it.

Similar to the role of *Ascl1* and Notch signalling in activating Müller cells, the Wnt/B-catenin signalling pathway also activates Müller glia and promotes proliferation by making them enter the cell cycle and promote the production of Müller glia progenitor cells (MGPCs) [51]. Interestingly, the activation of the Wnt pathway alone is sufficient to stimulate Müller glia to re-enter the cell cycle and proliferate [72]. The same study also showed that Wnt signalling is dependent on B-catenin signalling and that each pathway is individually capable of stimulating Müller cell proliferation. Unfortunately, the number of differentiating Müller cells in the above study was limited. However, another interesting study by Yao et al., 2018 showed that Müller glia were able to differentiate in vivo after the activation of the Wnt/B-catenin signalling pathway to stimulate cell proliferation. This was followed by the promotion of differentiation by the photoreceptor differentiation genes Otx2, Crx and Nrl in a mouse model of congenital blindness [71]. Another study by Angshumonik et al., 2016 revealed the relationship between Wnt/B-catenin and TGF-B signalling pathways in modulating Müller glia transdifferentiation into photoreceptors in vitro [52], which draws attention to the active role that the Wnt/B-Catenin might be acting on the regulation of Müller glia reprogramming in the mammalian retina.

Müller glia activation can occur by cell-to-cell fusion. For example, the activation of the Wnt/B-catenin signalling pathway occurred following the transplantation of iPSCs into NMDA-treated mouse retina. This coincided with the fusion of the cells with Müller, amacrine and ganglion cells that then dedifferentiated and regenerated the amacrine and ganglion cells [75]. In contrast, the downregulation of the Wnt/B-catenin signalling pathway (TGF-B signalling factors) resulted in the differentiation of Müller glial cell cultures with stem cell characteristics into photoreceptors [52].

When haematopoietic stem and progenitor cells (HSPS) were transplanted in the mouse retina, they fused only with Müller cells and the hybrid cells acquired stem cell characteristics and differentiated into photoreceptors [76]. The molecular mechanisms of the cell-to-cell fusion activation of Müller glia discussed are unknown and lack convincing in vivo functional tests. More research is required in this area to better understand the process and indeed to determine what purpose it might serve.

The role of the Sonic hedgehog (Shh) signalling pathway in Müller glia reprogramming has also been investigated. It has been suggested that the Shh pathway promotes Müller glia proliferation and stem cell characteristics in the injured rat retina [62]. Recently, the role of the Shh pathway in Müller glia transdifferentiation into photoreceptors was described [70]. Both Angbohang et al. (2016) and Gu et al. (2017) showed evidence of direct transdifferentiation of Müller glia into photoreceptors using different signalling pathways, Wnt/B-catenin and Shh, respectively, which opens up the possibility of modulating Müller glia reprogramming in retinal repair. However, both studies were in vitro; therefore, this transdifferentiation phenomenon in Müller glia still needs to be translated in vivo.

There is an argument that cellular metabolism, which refers to a set of chemical reactions in the cell that affect cell fate decisions, plays a role in Müller glia reprogramming. Although the transition from glycolysis, which stem cells use as a source of energy during pluripotency stages, to oxidative phosphorylation in the mitochondria during differentiation is well documented in the literature [77,78], its relationship with Müller cell reprogramming and differentiation is somehow indirect. However, cellular metabolism might still play a role in the molecular cascade during Müller cells’ differentiation, but whether that role is a cause or a consequence of differentiation is still unclear.

There are several other potential factors discussed in the literature on Müller glia reprogramming in mammals, including the regulation of the immune microenvironment, the different subtypes of Müller cells [16] and factors limiting Müller glia reprogramming in mammals, such as the complexity of the mammalian system and other evolutionary events that might halt the regenerative potential of Müller glia in mammals. Similar to the discussion about cellular metabolism, since it involves a multidisciplinary event, their direct involvement in Müller glia reprogramming is unclear. For example, the activation of the immune cells, such as microglia, upon injury is a systemic event and therefore it is hard to conclude its direct effect on the reprogramming of Müller glia.

The field of Müller glia-mediated retinal regeneration is very dynamic (fast-moving) and there has been a number of new developments in the field. Most of the recent developments were reported in either cold-blooded or warm-blooded vertebrates, some of which investigated molecular pathways that may improve the current understanding of the reprogramming of Müller glia in mammals. For example, a recent study by Palazzo et al. (2019) investigated the role of the Nuclear Factor Kappa B (NF-kB) pathway in Müller glia reprogramming in chick embryos, which showed that a component of the NF-kB pathway expressed by Müller cells, inhibits the reprogramming of Müller glia into retinal neurons [79], providing targets that can potentially be responsible for the lack of regeneration of mammalian Müller cells into retinal neurones.

## 5. Clinical Applications of Reprogrammed Mammalian Müller Glia

Clinical applications of reprogrammed Müller glia depend on the ability to successfully induce adult Müller glia to regenerate the retina. In mammalian systems, the regenerative potential of Müller glia is inefficient and fails to regenerate retinal neurones [49,69]. This makes it particularly difficult to study the mechanisms of mammalian reprogramming of Müller glia in vivo. Hence, there is a need for a model system to recapitulate retinal specific disease and study retinal development ex vivo. The reprogramming of adult somatic cells into induced pluripotent stem cells (iPSCs) has revolutionised regenerative medicine [80]. One promising approach is the reprogramming of adult Müller glia into iPSCs. Recently, a study by Slembrouck-Berc et al. demonstrated that Müller glia-derived iPSCs can be reprogrammed to acquire stem cell characteristics and differentiate into retinal pigment epithelial cells (RPE) and retinal organoids that harbour multipotent progenitors [81], which are able to differentiate into all major retinal neurones. Another study by Chung et al. and colleagues explored a new methodology of differentiating Müller glia from human embryonic stem cells (hESCs) by promoting the Notch signalling pathway [82]. Another example of understanding the mechanisms of Müller glia reprogramming is the use of ex vivo routes, which helps investigating the molecular queues behind a potential Müller glia regeneration in the mammalian systems. The development of ex vivo models not only helps understating the molecular mechanisms behind the lack of Müller glia regeneration in mammals, as the generation of differentiated Müller glial cells from iPSCs or hESCs can help future therapeutics such as gene editing or drug development using patient-derived glial cells.

In mammalian systems, differentiated retinal neurones do not have the ability to re-enter the cell cycle and divide like other cell types in the body. Injury to the retina that results in neuronal death often leads to vision loss due to the inability to replace lost neurones in the injured area. Other species, such as teleost fish (zebrafish), have a remarkable ability to regenerate the injured retina due to the efficient regeneration potential of their Müller glia [7,8,9]. Despite the considerable similarities between the mammal and zebrafish retina, the mammalian Müller glia fail to regenerate into retinal neurons in response to injury and their regeneration potential is restricted. Instead of dedifferentiating and initiating regeneration events, they experience reactive gliosis, which often involves cell proliferation and scar formation, failing to initiate regeneration [47]. The change in morphology and gene expression profile that the mammalian Müller glia undergo in response to injury fails to reprogram the cells into all major retinal neurons. Nevertheless, the ability of the mammalian Müller glia to upregulate genes associated with retinal stem cells and adopt stem cell characteristics [38,39,41] suggests that the mammalian Müller glia have the capacity to regenerate into retinal neurones under the right circumstances. Due to the quiescent nature of Müller cells, the process of activating them often involves retinal injury and the introduction of growth factors that stimulate differentiation. This injury certainly causes cell death and the release of apoptotic factors, which is perhaps unfavourable in mammalian systems. It is possible that future strategies to stimulate Müller glia in mammals involving no injury to an affected retina, such as introducing substances like glutamate [59], or activating signalling pathways that help their stimulation without causing injury such as Wnt or Notch signalling pathways [52,83], might help with initiating molecular cascades to stimulate regeneration.

## 6. Future Prospects

Retinal degeneration often results from cell death within the retina causing permanent sight loss and blindness. The idea of self-renewal potential within the retina is therapeutically appealing, especially with a transdifferentiation phenomenon to replace other cell types that die as a result of a disease. Müller glia possess a regeneration potential in fish in which the injured retina can self-repair. However, whilst the mammalian Müller glia can acquire stem cell characteristics, their regeneration potential is limited. Progress made in understanding the molecular mechanisms in the regeneration of Müller glia in fish has helped understand this lack of regeneration and repair in mammals. Further investigations will increase understanding of the mechanisms involved, including understanding the role of epigenetic modifications in Müller glia reprogramming.

Intrinsic mechanisms involved in regulating Müller glia’s response to injury might differ significantly between fish and mammals, but insights may help us exploit native mammalian Müller glia and encourage regeneration in favour of stem cell transplantation approaches. The transdifferentiation of Müller glia might not be an event solely initiated by one cell type; other cells might play a role by interacting with Müller glia and help to activate them, preparing the right environment and cellular events that modulate Müller glia reprogramming. Gaining more understanding of the regenerative potential of Müller glia in retinal repair will certainly be a game-changer in future therapeutic strategies in ocular disease.

## Figures and Tables

**Figure 1 cells-10-01957-f001:**
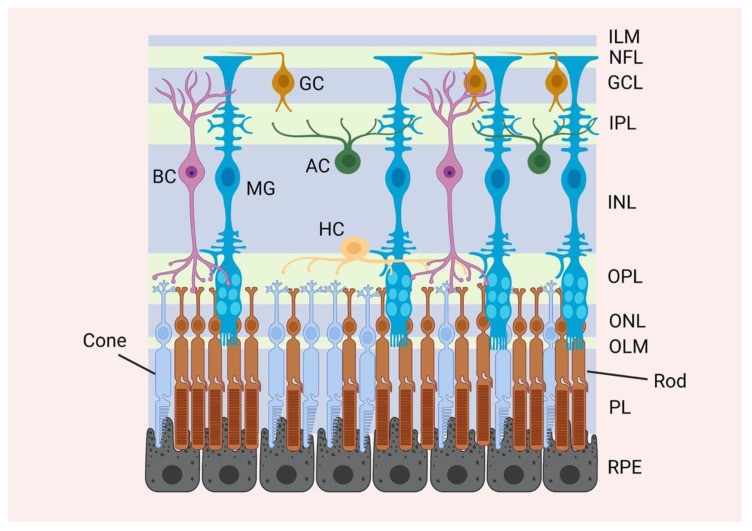
The mammalian retinal anatomy. A representative illustration of the major retinal layers and cell types. The mammalian retina is divided into 3 main laminar layers: the outer nuclear layer (ONL), the inner nuclear layer (INL) and the ganglion cell layer (GCL). There are six different retinal neuronal cell types and one glial cell type distributed within the nuclear layers: the rod and cone photoreceptor cell bodies are located in the ONL, whereas the cell bodies of the bipolar (BC), horizontal (HC), amacrine (AC) and Müller (MG) cells are located in the GCL. The cell bodies of the ganglion cells (GC) are located in the GCL. The processes of the different cells are extended into two plexiform layers. Processes from the photoreceptor cells are extended into the outer plexiform layer (OPL) to form synapses with the retinal neurones. Processes from the bipolar, horizontal, amacrine as well as Müller cells are extended into the inner plexiform layer (IPL). The axons of the ganglion cells are directed into the optic nerve through the nerve fibre layer (NFL). Müller glia span the length of the retina from the outer to the inner limiting membranes (OLM and ILM, respectively).

**Figure 2 cells-10-01957-f002:**
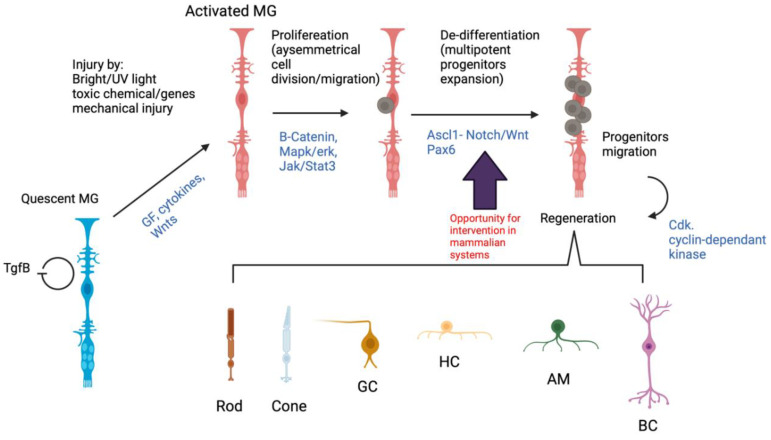
Müller glia regeneration-associated molecular cascades in zebrafish. A hypothetical pathway representing Müller glia activation in response to injury and the sequence of events that leads to retinal regeneration in zebrafish (black text). Müller glia are quiescent in absence of injury, a process regulated by TgfB/Smad pathway. Upon injury, Müller glia are encouraged start reprogramming and enter the cell cycle in a complex of events involving B-catenin, Mapk/erk and Jak/Stat3 signalling pathways, which stimulate proliferation. These pathways stimulate injury-dependant Ascl-1 expression, which regulates the generation of Müller glia-l-derived progenitor cells (MGPCs). Those sequences of events also occur in mammalian systems. However, the MGPCs in zebrafish retina expand and migrate to regenerate the injured retina. This regeneration potential is inefficient in mammalian systems. Intervention at key steps in the zebrafish Müller glia reprogramming might stimulate the regeneration potential in mammalian Müller glia (purple arrow).

**Table 1 cells-10-01957-t001:** Factors affecting Müller glia regeneration in mammals.

Signalling Pathway/ Factor	Function	Species Tested	Effect on Müller Glia Regeneration	References
*Ascl1*	Transcription factor	Birds, rodents	Stimulates	[49,55,56]
BMP-SMAD	Signalling pathways activated by a secreted factor (BMP) when binding to a transcription factor (SMAD)	Rodents	Stimulates	[56]
EGF/EGFR	Signalling pathways activated by a secreted factor (EGF) when binding to its receptor (EGFR)	Rodents	Stimulates	[49,57,58]
Glutamate	Neurotransmitter	Rodents	Stimulates	[59]
FGF2-FGFR-MAPK	Secreted factor (FGF2) binding to its receptor (FGFR) to activate a signalling pathway (MAPK)	Birds, rodents	Stimulates	[57,65,66,67,68]
Delta-Notch	Signalling cascade activated by a transmembrane ligand (Delta) binding to its receptor	Birds, rodents	Stimulates	[20,49,65]
Pax6	Transcription factor	Birds, rodents	Unknown	[49,69]
Insulin-IGF1-PI3K	Secreted factor (FGF2) binding to its receptor (FGFR) to activate a signalling pathway (MAPK)	Birds, rodents	Stimulates	[66,67,68]
SHH	Secreted factor	Rodents	Stimulates	[62,70]
TGF-B	Secreted factor	Rodents	Inhibits	[52,57,58]
Wnt/B-catenin	Signal transduction pathway	Rodents	Stimulates	[51,52,71,72]

## Data Availability

Not applicable.

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
