# Peer review of "Insights on the Regeneration Potential of Müller Glia in the Mammalian Retina"

_cells, 2021, doi:10.3390/cells10081957_

Round 1

Reviewer 1 Report

The review by Dr. MacLaren et al is an interesting and well organized paper and it summarized the most recent progress in the reprogramming of Muller cells and the regeneration of retina. Although there are several similar published reviewes , this review focused on the species differences of muller reprogarmming between zabra fish and mamals and gave some prospects in future clinical use. I suggest the author summarize the low efficiency and impairment of reprogrammed muller cells in the injuried or degenerative retina and gave some advice on how to improve it.

Author Response

Thanks for the comment, I have discussed the low efficiency of regeneration in the injured mammalian retina and proposed a possible strategy to overcome it in lines 333-355 (highlighted in yellow).

Reviewer 2 Report

This review by A. Salman et al on the regeneration of retinal neurons from MG provides a general overview of the field, this is an important area of retinal pathobiology. The review is very well-written and the illustration is general good. As this is a very dynamic (fast moving) field, it was a little disappointing for this reviewer, as there are very few cited research reports that could provide the new mechanisms for MG-mediated retinal regeneration were published within the the last two years. The review would be more attractive if the authors could discuss more new developments in the field, as claimed in the abstract. The following are a list of minor issues:

  1. Missing key words
  2. Fig 2 image is not of high quality.
  3. Line 253: Yao et al. (2018), not the right format.
  4. Missing author contributions.

Author Response

Thanks for the comment. I have already included recent research reports in the “Current understanding of the regeneration potential in of Müller glia in mammals” which are highlighted in yellow (lines 227-233). I have also discussed more recent developments on the use of iPSCs and hESCs in the clinical applications section (highlighted in the manuscript lines 317-328). I have also added discussion on other recent developments in the field in lines 301-310. I will also be happy to discuss any other reports that the reviewer feels important.

As for the list of minor issue, here is a point-by-point response to each issue. The changes have been made to the manuscript are highlighted in yellow

  1. Missing key words

My apology, I have originally included key words but they were missed when the manuscript was uploaded into the journal’s template. I have added the ley words in the appropriate section in the manuscript.

  1. Fig 2 image is not of high quality.

Thanks for the comment. The figure was created using a software and stored as a tif file at the highest quality possible. I have re-apploaded it and tried to optimise the image quality after importing from the software.

  1. Line 253: Yao et al. (2018), not the right format.

Thanks for the comment. The citation has been changed to the correct format.

  1. Missing author contributions.

Thanks for the comment. Author contributions were added originally but missed while transferring the manuscript to the journal’s template. I have included author contributions.

Please see attached file for point-by-point response.

Round 2

Reviewer 2 Report

The authors addressed my concerns.